# Reducing Construction Dust Pollution by Planning Construction Site Layout

Guowu Tao [1], Jingchun Feng [1,*], Haibo Feng [2], Hui Feng [1] and Ke Zhang [1]

[1] Business School, Hohai University, Nanjing 211100, China; taoguowu@hhu.edu.cn (G.T.); huifeng@hhu.edu.cn (H.F.); kezhang@hhu.edu.cn (K.Z.)

[2] Department of Mechanical and Construction Engineering, Northumbria University, Newcastle NE2 1XE, UK; haibo.feng@northumbria.ac.uk

[*] Correspondence: jcfeng@hhu.edu.cn

**Abstract:** Many construction activities generate fine particles and severely threaten the physical health of construction workers. Although many dust control measures are implemented in the industry, the occupational health risks still exist. In order to improve the occupational health level, this study proposes a new method of reducing the construction dust pollution through a reasonable site layout plan. This method is based on the field measurement and dust diffusion law. The dust diffusion law can be fitted based on the field monitoring data. With diffusion law, the average dust concentration exposed to workers of different site layouts can be simulated. In addition, the cost of the dust control method is a concern for site managers. Therefore, the total transportation cost reduction is another optimization objective. Finally, the multi-objective particle swarm optimization (MOPSO) algorithm is used to search for an optimized site layout that can reduce dust pollution and transportation cost simultaneously. The result shows that average dust concentration exposed to workers and total transportation cost are significantly reduced by 60.62% and 44.3%, respectively. This paper quantifies the construction dust pollution and provides site managers with a practical solution to reduce the construction dust pollution at low cost.

**Keywords:** construction dust; construction site layout planning (CSLP); occupational health; MOPSO; BIM

## 1. Introduction

With rapid economic and industry development, air pollution has become a serious environment problem worldwide, especially in urban areas [1]. Air pollution causes critical respiratory illness to human beings. Inhalation of particulate matter (PM) suspended in the air can cause various long-term respiratory diseases and premature deaths [2]. Therefore, people are gradually raising attention to air pollution in these years. The construction industry is one of the primary air pollution sources in most counties [3]. Additionally, the construction industry is a labor-intensive industry and many workers have to work very close to various dust sources. Many construction activities, such as cement mixing and template cutting, can produce much dust during construction phases [3,4]. These dust pollutes the surrounding environment [5] and also threatens the physical health of construction workers [6,7]. Many researchers have found that this kind of working environment can lead to an increase in the number of diseases related to cardiovascular, respiratory and the skin [8–10]. Therefore, construction dust's negative effect on workers has attracted the attention of whole construction industry and academia. It is necessary to take some measures to protect construction workers.

Many researchers have conducted research about the health impact of the PM exposure on workers [3,11–14]. There are three main research directions: health impact assessment, dust monitoring, and dust control and prevention. The first one is the health impact assessment. Some researchers collected related medical data for years to confirm that

the construction dust has a negative impact on construction workers. Bergdahl et al. [11] followed 317,629 construction male construction workers for 28 years and tried to find out the association between mortality from chronic obstructive pulmonary disease (COPD) and occupational exposure to construction dust. The final results show that there is increased mortality from COPD among those exposed to the construction dust, especially among never-smokers. Borup et al. [12] also analyzed the records and validated the high association of COPD among construction workers. Those with pre-existing cardiopulmonary problems and elderly workers are more sensitive to the PM [15]. PM2.5 decreased the average life span by 8.6 months in EU countries [16]. In general, construction dust threatens the physical health of workers severely.

Based on the common understanding of health risks caused by construction dust, in recent years, many researchers have tried to quantificationally evaluate the dust impact from construction sites. Tong et al. [4] applied Monte Carlo simulation and the United States Environmental Protection Agency (USEPA) risk assessment model to explore the health effects of construction dust on practitioners in the construction industry. They also divided the construction site into different functional zones, such as steel zone and floor zone, and evaluated the health impact of workers in different zones separately. The disability-adjusted life year (DALY) parameter, which was developed by Murry at Harvard University and World Health Organization (WHO) [17], was widely used in the health risk assessment study. Researchers applied DALY to quantitatively measure the burden of disease and perform a quantitative assessment of health damage [4]. Based on the DALY parameter, some researchers have tried to translate the health risks into the economic significance through willingness to pay (WTP) [13,14]. These studies analyzed the impacts of the construction dust on people and identified the main influencing factors, such as exposure time, exposure methods, dust concentration, etc. With these research foundations, researchers built the relationship between construction dust and occupational health risk. In most studies, the concentration of dust that workers are exposed to is a very important factor. It is highly positively correlated with occupational health risk. Therefore, we improve the worker's occupational health by reducing the dust concentration that workers are exposed to in this study.

Another main research direction in this research field is dust monitoring. How to monitor the dust concentration accurately is always a complex issue. Many researchers have promoted the dust monitoring research from multiple aspects [5,18,19]. In order to obtain accurate monitoring results, the up–down wind direction method was used to eliminate the interference due to environment background and reveal the absolute value of incremental dust concentration from construction sites [5,18,20]. In the process of dust monitoring, some researchers found that some of the environmental factors, such as wind speed [21], humidity [22] and temperature [23] affect the accuracy of dust sensor. In addition, researchers found that construction activities also affect the dust concentration [3–5,18]. Li et al. [3] found that the top three respirable exposures are from cement mixing, concrete breaking and manual demolition. Yan et al. [5] analyzed the field monitoring data and found that the construction vehicles were one of main influencing factors of construction dust. Tong et al. [4] compared the dust concentration in different construction zones and the results indicated that the template zone is the most polluted area. Except the research above, during the dust monitoring process, some researchers also analyzed the chemical composition of the dust [19,24] and diffusion law of dust from construction sites [18,20].

As for the dust monitoring equipment, many modern-day devices, such as sensors and the wireless sensor network (WSN) have been applied into practice nowadays with the advancement of technology. In the past, air quality monitoring stations were often built to collect dust concentration data. However, these stations have many limitations, such as the large size, high cost and power requirement [25]. To overcome these barriers, some researchers compared different sensors and applied these low-cost sensors and WSN to monitor the dust concentration on construction sites. Budde et al. [26] suggested

that air quality monitors could be replaced with low-cost sensors due to their ease of use. Cheriyan and Choi [27] used low-cost sensors, Alphasense and Sharp sensors to monitor PM, and the results shows that these sensors have good performance during the monitoring process.

The final main research direction is dust control and prevention. The research above confirmed that the dust generated from construction pollutes the air and threatens the physical health of workers. The PM10 and smaller dust particles can reach the respiratory tract and lung, which can result in an irreversible respiratory disease called pneumoconiosis [3,15]. Therefore, many researchers and practitioners have put lots of effort into dust control and prevention research. Some studies used technical measures to control the construction dust. Li et al. [3] analyzed 783 samples in Hongkong and summarized that there are three commonly used dust control measures: local exhaust ventilation (LEV), blower fans and wet methods. Respirators and wet methods are the most widely used protective measures in the Hong Kong construction industry. Chen et al. [13] evaluated the isolation effect of dust masks and the results showed that the health risk could be reduced by 26% under the actual effect. Tjoe Nij et al. [28] proposed that only the combined use of more than one control measure can reduce the construction dust's negative effect to acceptable levels. In addition to technical measures, reasonable managerial instruments or measures are vital as well. Wu et al. [29] proposed that the formulation of targeted regulation and establishment of an appropriate charging scheme could increase contractors' willingness in mitigating construction dust. They also found that it is essential to give construction workers specific dissemination and training to increase their environmental awareness. In addition, a corresponding monitoring system could guarantee the effectiveness of the regulation and charging scheme.

It can be seen that researchers and practitioners have made many efforts in dust pollution control, and many dust control measures have already been taken in the construction industry. However, the current air condition on the construction sites is still not fully satisfactory [29]. These control measures have some obvious shortcomings. The wet methods, such as the spraying system and manual watering, need to continuously spray clean water mist into the air during the construction period. Considering that most of the construction projects have a long period, it would consume much of the water resources. This may lead to a local water shortage, especially in relatively arid areas. In addition, in order to ensure the safety of the use of electrical equipment, it is required to keep a safe distance from the spraying system or to install additional waterproof covers. The dust masks can significantly reduce the health risk of workers, but the uneven quality and the inappropriate usage of masks would reduce their isolation effect. The actual isolation effect of dust mask is only 26% in the construction industry [13]. Other measures, such as LEV and blower fans, would require much electricity to operate during construction. Therefore, there is an urgent need for an economical, convenient and practical dust control solution in the construction industry.

This study proposes a new method that uses a reasonable site layout plan to reduce the average dust concentration exposed to workers. This method allocates the facilities on sites according to the dust diffusion law and number of workers on each working zones. By increasing the distance between dust pollution sources and workers, this method can reduce the average dust concentration exposed to workers. Thus, the negative effect of construction dust could be reduced. However, the longer distance between facilities would increase the transportation cost on the site. Therefore, the multi-objective particle swarm optimization (MOPSO) algorithm is applied in this study to balance the construction dust negative effect and total transportation cost. In addition, the BIM model, due to its rich sources of building information, is also used in this study to automatically provide basic construction information [30], which can help site managers save much time and effort in designing the site layout plan in the pre-construction stage.

## 2. Methodology

In order to properly design the site layout to reduce the negative dust impact while maintaining the minimum on-site transportation cost, the following framework is proposed, shown in Figure 1.

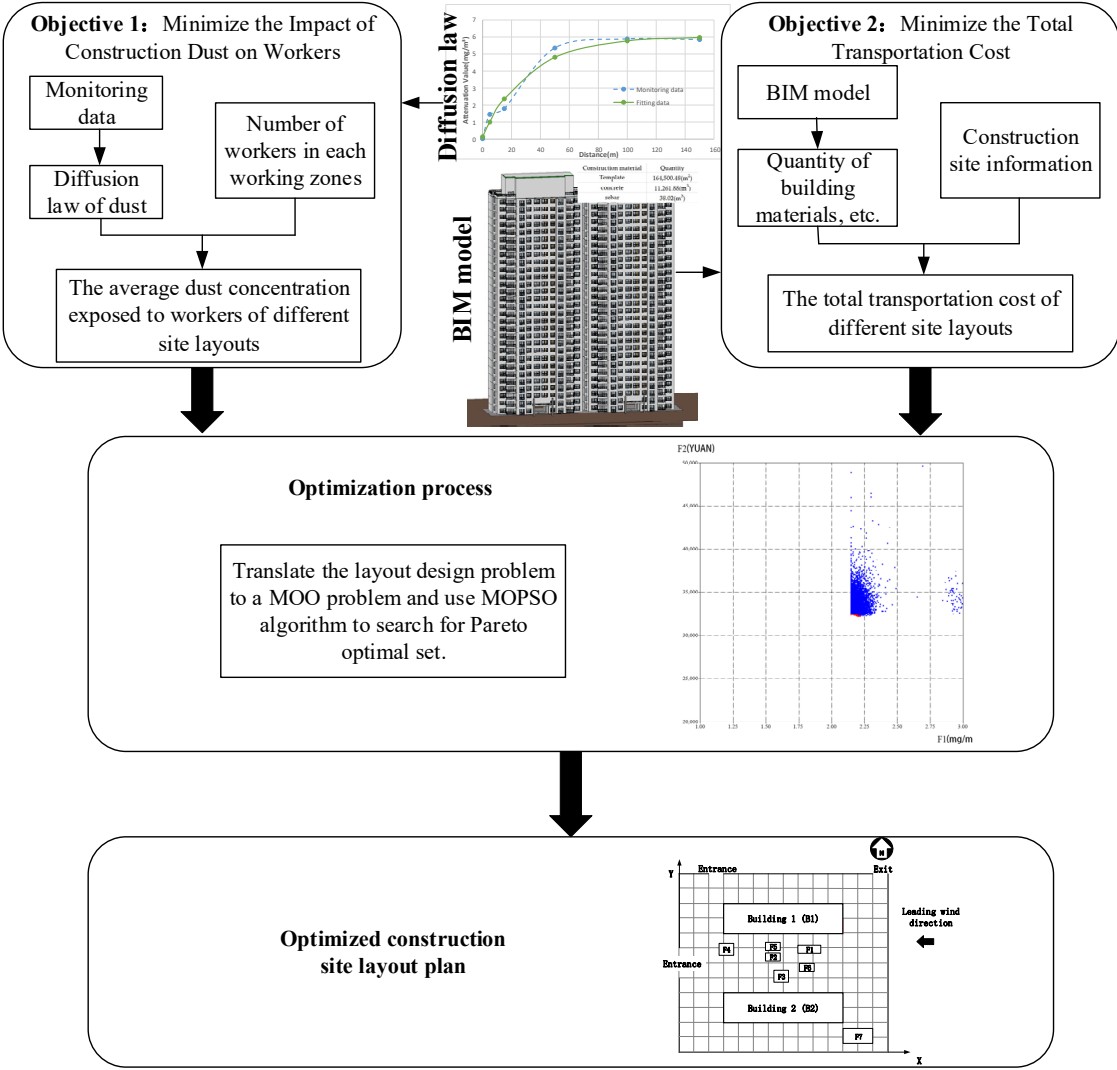

**Figure 1.** The framework for reducing construction dust effect by CSLP.

### 2.1. Determine the Scope of Study

In this study, reducing the impact of construction dust on workers and the on-site transportation cost of building materials are the two objectives. As for reducing the impact of construction dust, this study only focuses on the impact of dust generated by construction activities on the construction site. Dust from outside the construction site is not considered in this study. The construction site condition (e.g., the phase of construction and wind condition of the construction site) and number of workers in each facility are also considered in this study. In addition, the proposed model only considers the direct impact of construction dust on workers. The indirect impact caused by physicochemical effects is not considered in this study [18]. As for the total on-site transportation cost, only the horizontal transportation of building materials within the boundary of the construction site is considered in this study. The total transportation cost consists of the horizontal and vertical transportation costs. The horizontal transportation cost generally accounts for a large proportion of the total transportation cost. The vertical transportation of building materials generally relies on tower cranes, which cost much less than the horizontal one [31].

Therefore, only the horizontal transportation cost is applied when optimizing the site layout. Generally, workers would engage their corresponding types of construction activities during the construction period, and they spend much time at their workplace. Therefore, the movement of workers is ignored in this study.

### 2.2. Modeling of Multi-Objective Construction Site Layout Problems

The purpose of the study is to reduce the impact of construction dust on workers and total on-site transportation cost by planning the construction site layout. In this section, a multi-objective optimization (MOO) model is built to balance the construction dust impact and total transportation cost.

### 2.2.1. Minimize the Impact of Construction Dust on Workers

The construction dust can lead to construction workers suffering from respiratory diseases, such as cardiovascular disease, cerebrovascular disease, etc. [32]. In a previous study, researchers often took the inhalation health risk assessment model [33], which is recommended by the United States Environmental Protection Agency (USEPA), to evaluate the health damage of workers. According to the USEPA model, the degree of health damage is highly positively correlated with the dust concentration. In other words, the prevalence will increase with an increasing construction dust concentration. Therefore, the concentration of dust is selected as the indicator to measure the impact of construction dust on workers. In this study, the total suspended particulate matter (TSP) concentration in the air is chosen as the indicator to represent the dust concentration on the construction sites. The TSP denotes the total suspended particulates with aerodynamic diameters of less than 100 μm, which could cause a threat to workers' health [4]. To eliminate the interference due to environment background dust and accurately reflect the impact of construction dust on workers, the up–down wind direction method is adopted in this study to monitor the dust concentration [5,18,20]. The leading winds, which are the prevailing winds in the meteorology, blow constantly in a given direction [34]. Therefore, the leading wind direction is applied as the up–down wind direction in this study [5,18].

Generally, the pollutant is subjected to diffusion in all directions in the presence of advection and settling due to gravity [35]. Therefore, under the action of wind, the attenuation value of dust concentration shows a certain relationship with the distance from the pollution source. The construction dust attenuation relationship between construction dust concentration and distance in the wind direction can be statistically fitted depending on the monitoring data and exponential law model [18,36] (see Figure 2). For the construction dust concentration at the workers' workplaces $i$ ($i = 1, 2, \ldots, m$) that originates from the dust source $j$ ($j = 1, 2, \ldots, n$), the construction dust concentration is the sum of each dust source, which can be denoted by Equations (1)–(3):

$$CDC_i = \sum_{j=1}^{n} C_{ij} + C_b, \tag{1}$$

$$C_{ij} = \begin{cases} SC_j - Y, & \left| SC_j \geq Y \right. \\ 0, & \left| SC_j < Y \right. \end{cases}, \tag{2}$$

$$Y = 6.02 - 5.882e^{-\frac{wd_{ij}}{31.59}}. \tag{3}$$

In Equation (1), $CDC$ is the construction dust concentration at workers' workplace $i$, where $i = 1, 2, \ldots m$; $C_{ij}$ represents the construction dust concentration at workers' workplace originating from the dust source $j$, which can be derived from Equation (2); $C_b$ is the background construction dust concentration. Due to the up–down wind direction method [20], the upwind point concentration is taken as $C_b$.

In Equation (2), $SC_j$ is the concentration of the dust source $j$; $Y$ is the construction dust concentration attenuation value. If the concentration of dust source $SC_j$ is greater than the attenuation value $Y$, then $C_{ij}$ is equal to the concentration of dust source $SC_j$ minus attenuation value $Y$. If the concentration of dust source $SC_j$ is less than the attenuation

value $Y$, then the $C_{ij}$ is equal to 0, i.e., in this situation, the dust source $j$ has no negative impact on the workplace $i$. In Equation (3), $wd_{ij}$ is the distance between workplace $i$ and dust source $j$ in the leading wind direction. This equation is derived by fitting the monitoring data (Figure 2).

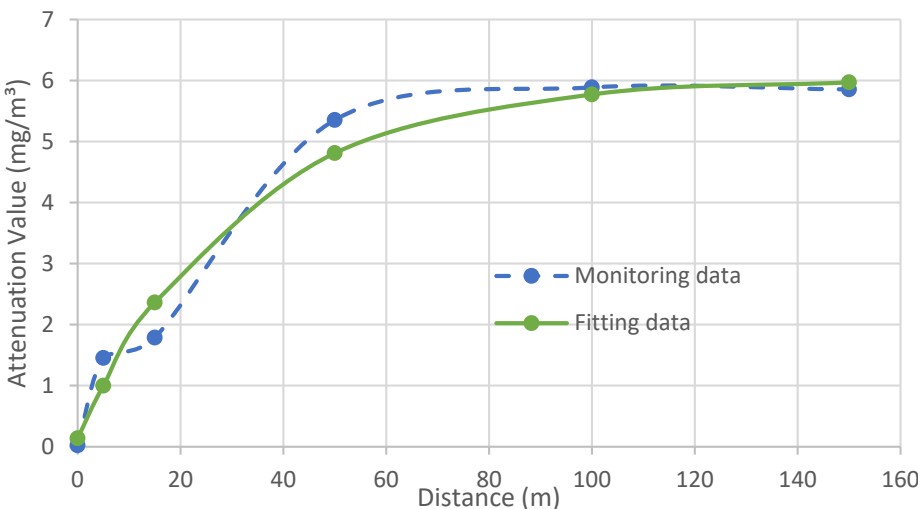

**Figure 2.** The construction dust concentration attenuation relationship.

In order to reduce the impact of construction dust on workers, the construction dust reduction objective function $F_1$ should fulfill the following requirement in Equation (4):

$$F_1 = \min \frac{1}{W} \sum_{i=1}^{m} CDC_i \times W_i. \tag{4}$$

$F_1$ is the objective function that minimizes the average construction dust concentration exposed to workers. $W$ is the number of workers on the construction site and $W_i$ is the number of workers at the workplace $i$. The number of workers on the construction site is equal to the sum of number of workers at all workplaces, i.e., $W = \sum_{i=1}^{m} W_i$. In order to reduce the impact of construction dust, the main dust sources should be assigned as far as possible at the downwind points.

### 2.2.2. Minimize the Total Transportation Cost

A good construction site layout plan could help site managers reduce the impact of construction dust within the cost budget [37]. On the construction sites, the transportation of building materials incurs a certain cost. The total transportation cost is an important part of the cost budget. Therefore, the second optimization objective, $F_2$, is to minimize the total transportation cost, as shown in Equation (5).

$$F_2 = \min \sum_{i=1}^{m-1} \sum_{j=i+1}^{m} d_{ij} C_{ij} f_{ij}. \tag{5}$$

In Equation (5), $d_{ij}$, $C_{ij}$ and $f_{ij}$ denote the Euclidean distance, the transportation cost per unit length and the frequency of transportation between facilities $i$ and $j$. Different site layouts would profoundly affect the transportation distance $d_{ij}$ between facilities. $C_{ij}$ depends on the mode of transportation. $f_{ij}$ is determined by the quantity of building materials to be transported and transportation mode.

### 2.3. Constraints of the Construction Site Layout Problem

When optimizing the construction site layout planning, the proposed model should comply with the corresponding constraints to make the optimization results more realistic. For the convenience of representation and calculation, the construction site layout should be put in the coordinate system.

### 2.3.1. Construction Site Boundary Constraint

All the facilities should be assigned within the boundary of the construction site. To prevent facilities positioned outside the borders of the available locations, Equations (6)–(9) are formulated:

$$x_i \geq \frac{b_i}{2}, \tag{6}$$

$$x_{bd} - x_i \geq \frac{b_i}{2}, \tag{7}$$

$$y_i \geq \frac{l_i}{2}, \tag{8}$$

$$y_{bd} - y_i \geq \frac{l_i}{2}. \tag{9}$$

where $(x_i, y_i)$ denotes the Cartesian coordinates of the centroid of facility $i$; $b_i$ and $l_i$ represent the horizontal and vertical lengths of facility $i$, respectively; and $x_{bd}$ and $y_{bd}$ mean the horizontal and vertical boundaries of the available locations. The four equations above ensure that the boundaries of each facility are all within the borders of the construction site.

### 2.3.2. Overlapping and Safety Constraint

When two or more facilities are assigned on the construction site in the same phase, facilities cannot overlap each other or overlap with buildings. In addition, the minimum distance between the facilities is set in this model to avoid mutual interference and ensure the construction safety. The constraint is applied using Equation (10):

$$\min\{0.5(b_i + b_j) + h_{ij} - |x_i - x_j|, 0.5(l_i + l_j) + v_{ij} - |y_i - y_j|\} \leq 0. \tag{10}$$

where $h_{ij}$ and $v_{ij}$ denote the minimum horizontal and vertical safe distances between facilities $i$ and $j$, respectively. These two parameters can ensure the safety of construction activities in the corresponding facility and basic transportation of building materials on the sites. The specific values of the minimum safety distance can be determined according to the requirements of construction projects.

### 2.4. Optimization Using MOPSO Algorithm

The proposed model above reflects how the construction site layout affects the two objectives: impact of construction dust and total transportation cost. From the computation point of view, the CSLP is an NP-hard question [38]. In addition, the proposed model in this study is also a multi-objective optimization (MOO) problem [37]. According to several research studies [37,39,40], heuristics algorithms are often used to solve CSLP. MOPSO, due to its ease of implementation and ability to handle multi objectives [41], is applied in this study to solve the site layout problem. The essence of MOPSO is that the local and global optimal values guide the particles and generate the brand-new position and speed of the particle. This can make the result of the proposed model converge toward the global optimal position. Due to these characteristics of the MOPSO algorithm, the construction dust pollution and total transportation costs can be reduced simultaneously. The details of the algorithm are as follows.

#### Procedure of MOPSO Algorithm

In this algorithm, the coordinates $(x_i, y_i)$ of the centroids of facility $i$ are regarded as a decision variable. A set of coordinates, which represent all available positions in the construction site, is regarded as the input of the algorithm. The detailed procedure of the MOPSO is the following.

Step 1. Initialize the particles. Randomly generate initial position $POP_i(0)$ and velocity $V_i(0)$ for each particle. In the construction site layout planning, a possible position of the particle represents a possible assignment for all site facilities.

Step 2. Initialize the external archive and find the initial previous best position of particle $i$ ($pbest_i$) and the global best position of particles ($gbest$). The fitness function of each particle is calculated in terms of the objective function. Compare the fitness function value of each particle and store the nondominated particles in the external archive. The global best position ($gbest$) is a value taken from the external archive. Since the particles in the archive are all nondominated, the global best position should be selected according to certain rules. The details are as follows.

(a) Divide the objective function space into grids. The objective function space is a coordinate system and particles can be located in this system according to the values of the particle's objective functions, $F_1$ and $F_2$.

(b) Choose a grid by using Roulette-Wheel Selection. The grid which contains more than one particle is assigned a fitness equal to the result of dividing any number $x > 1$ by the number of particles that the grid contains. This kind of fitness assignment can ensure that the grid which has fewer particles has more probability to be selected. Then Roulette-Wheel Selection is applied to choose a grid. This selection method can avoid the particles from quickly converging to the local optimum position.

(c) Randomly select a particle within the grid chosen above as the global best position $gbest$. The initial previous best particle $i$ is the initial position itself (see Equation (11)).

$$pbest_i = POP_i. \tag{11}$$

Step 3. Update the velocity and position of particles. The velocity $V_i(t)$ is updated according to Equation (12):

$$V_i(t+1) = w \times V_i(t) + c_1 r_1 \times (pbest_i(t) - POP_i(t)) + c_2 r_2 \times (gbest_i(t) - POP_i(t)). \tag{12}$$

In Equation (12), $w$ is the inertia factor; $c_1$ and $c_2$ are the local and global acceleration coefficients, respectively; $r_1$ and $r_2$ are random numbers uniformly distributed within [0,1]; and $t$ represents the $t$th iteration of the algorithm. It can be seen from Equation (12) that the velocity is guided by the previous best position and global best position simultaneously.

The position of particles $POP_i$ is updated as follows:

$$POP_i(t+1) = POP_i(t) + V_i(t+1). \tag{13}$$

The updated position is equal to the position of particle in the last iteration plus the updated velocity.

In order to avoid the particles from searching beyond the available search space, all the particles need to be checked. If the particle goes beyond the boundary, the position of the particle takes the value of the corresponding boundary, and the velocity is multiplied by $-1$ so that the particles can search in the opposite direction.

Step 4. Update the fitness function value of the particles and external archive. Calculate the fitness value according to the position of the particles in the $(t+1)$ iteration. Then add the nondominated particles into the external archive and eliminate the dominated particles from the archive. Due to the limited size of the archive, particles which are located in more populated area of objective function space are more likely to be eliminated when the external archive is full.

Step 5. Update the previous best position and global best position of particles. When the current position of particle is better than previous best position $pbest_i$ in the last iteration, update the $pbest_i$ with the current position.

$$pbest_i = POP_i \tag{14}$$

If the current position is worse, then the value of $pbest_i$ is kept in its current iteration. If neither of them is dominated by the other, one of them is selected randomly. As for the $gbest$, it is updated in the same way as that in step 2.

Step 6. Repeat steps 3 to 5 until the number of iterations reaches the set value. In the external archive, the position of particles represents the location of the facilities on the construction site. The objective functions of particles in the archive represent the impact of construction dust on workers and the total transportation cost.

## 3. Case Study

In order to validate the effectiveness and practicability of the proposed methodology above, a residence construction project is adopted. In this section, an analysis for the construction project and a comparison with the original site layout plan are conducted to highlight the approach.

### 3.1. Case Study Description

The residence construction project is located in Nanjing, China. Usually, the dimensions of the temporary facilities are determined according to the scale of the construction projects and the amount of building materials. In this case, the dimensions of facilities, which are all predefined by site managers, are presented in Table 1. The facilities on the site could be divided into three categories: (1) storage facilities, (2) processing facilities and (3) residence facilities. Building materials are delivered to the construction site and then stored temporarily in storage facilities. When needed, building materials are first transported to the processing facilities for processing and then transported to the floor zone for construction. As for the residence facility, it provides a comfortable office and rest environment for site managers and workers. This residence construction project uses BIM to design and construct. Therefore, the quantity of different building materials could be extracted from the BIM model (see Figure 3). Additionally, trunks and forklifts are used to transport building materials horizontally in this construction project.

**Table 1.** The required facilities and their dimensions.

| Symbol | Facility | Type | Width in $x$ Direction (m) | Length in $y$ Direction (m) |
|--------|----------|------|---------------------------|----------------------------|
| F1 | Template storage yard | Storage | 15 | 5 |
| F2 | Rebar storage yard | Storage | 10 | 5 |
| F3 | Concrete warehouse | Storage | 10 | 8 |
| F4 | Template processing yard | Processing | 10 | 8 |
| F5 | Rebar processing yard | Processing | 10 | 5 |
| F6 | Concrete mixing station | Processing | 10 | 5 |
| F7 | Site office | Residence | 20 | 10 |

According to the climate data, the leading wind direction of the construction site is east. That means that the construction dust sources located in the east side will affect the workers on the west side of construction site. Due to the long construction period, many workers and many kinds of construction activities required, the superstructure construction stage was selected in this case study. The number of workers at different workplaces in this residence construction project can be seen in Table 2.

**Table 2.** Number of workers at the corresponding workplaces.

| Symbol | Workspace | Number of Workers Required |
|--------|-----------|----------------------------|
| F4 | Template processing yard | 5 |
| F5 | Rebar processing yard | 10 |
| F6 | Concrete mixing station | 5 |
| F7 | Site office | 10 |
| B1/B2 | Floor zone | 20 [1] |

[1] Each floor zone needs 20 workers.

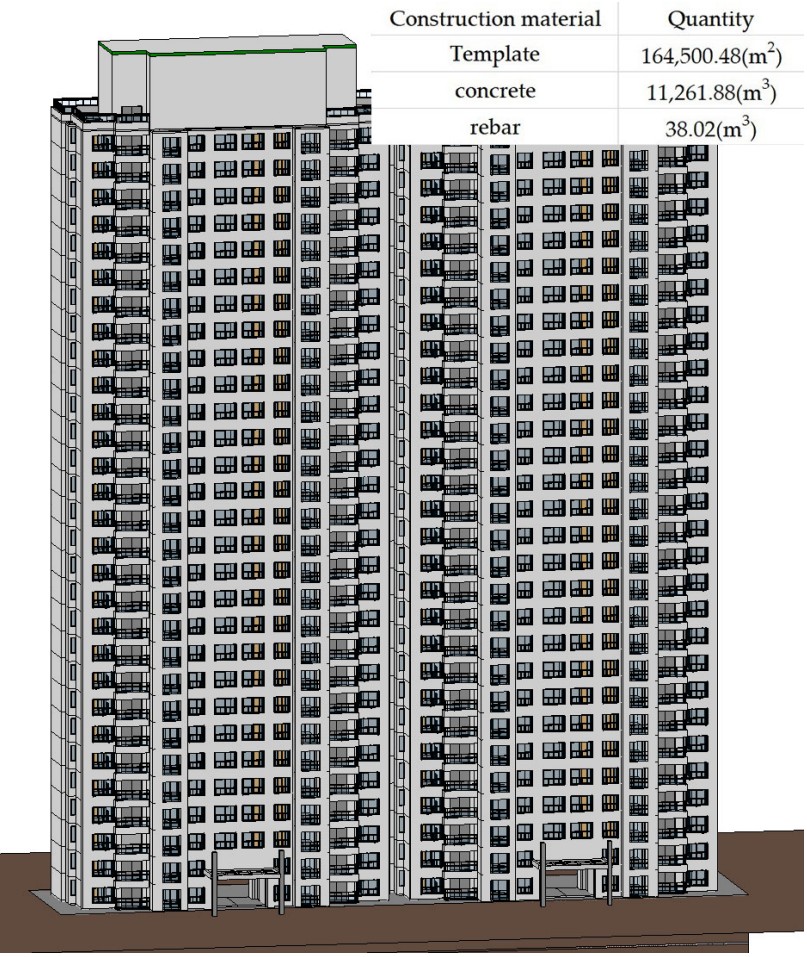

| Construction material | Quantity |
|---|---|
| Template | 164,500.48(m²) |
| concrete | 11,261.88(m³) |
| rebar | 38.02(m³) |

**Figure 3.** BIM model of the project and corresponding quantity of building materials.

On the construction site, some construction activities, such as template processing, rebar processing, cement mixing, etc., generate lots of construction dust [3,4]. In this construction project, the TSP monitoring tool, dust sampler, was applied to collect dust concentration data of major construction dust sources. In order to measure the dust concentration more accurately, the dust sampler was mounted on a tripod and sampling points were close to the operator without affecting the operator. In addition, the wind speed of the construction site is around 1.0 m/s, which is a light wind, during the monitoring period. The detailed concentration of dust sources is seen in Table 3.

**Table 3.** Mean concentration of dust sources.

| Symbol | Dust Source | Corresponding Construction Activity | Dust Concentration (mg/m³) |
|---|---|---|---|
| F4 | Template processing yard | Template processing | 5.65 |
| F5 | Rebar processing yard | Rebar cutting and bending | 1.50 |
| F6 | Concrete mixing station | Concrete mixing | 2.24 |
| B1/B2 | Floor zone | Concrete pouring, template dismantling, etc. | 1.20 |

### 3.2. Results of the Case Study

The optimized construction site layout plans could be generated by applying the proposed multi-objective construction site layout model and the MOPSO algorithm. Python was used to code the model and algorithm. After a few rounds of searching, the Pareto front and Pareto optimal set, which represents potential construction site layout plans, was

generated. In Figure 4, the red dots represent the non-dominated solutions and the blue dots represent the non-optimal solutions. One characteristic of the non-dominated solution is that a gain in an objective from one solution to the other is only obtained by sacrificing at least one other objective [42]. Due to this characteristic, the site managers need to choose the most suitable solution according to the practical situation of the construction project. In this case, the site managers pay lots of attention to the health of construction workers. Therefore, the red dots on the left are more likely to be selected by managers. However, the total transportation cost of the leftmost red dot is particularly high, compared with other dots in Pareto front. Finally, the dot enclosed by the red box is selected as the final optimization solution. The optimized construction site layout plan is shown in Figure 5.

**Figure 4.** Pareto front for the proposed two objective functions.

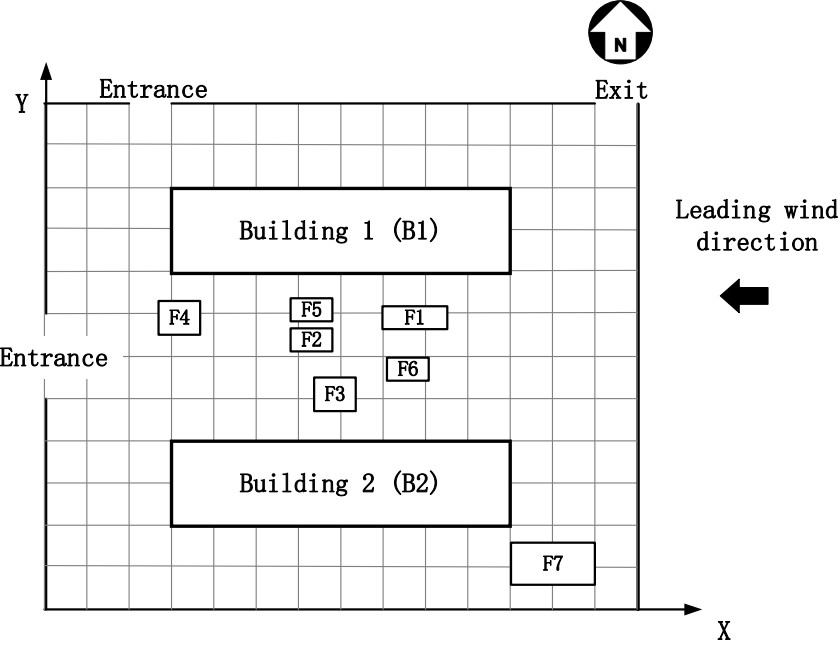

**Figure 5.** The optimized construction site layout plan.

In order to demonstrate the benefit of the proposed method more intuitively, the original construction site layout plan (see Figure 6), made by site managers, was also considered in this study. Traditionally, the site layout plans are often made based on site managers' experience. Table 4 compares the impact of construction dust and total transportation cost corresponding to the two cases. From Table 4, we can see that optimized CSLP plan has better performance in both construction dust control and transportation cost.

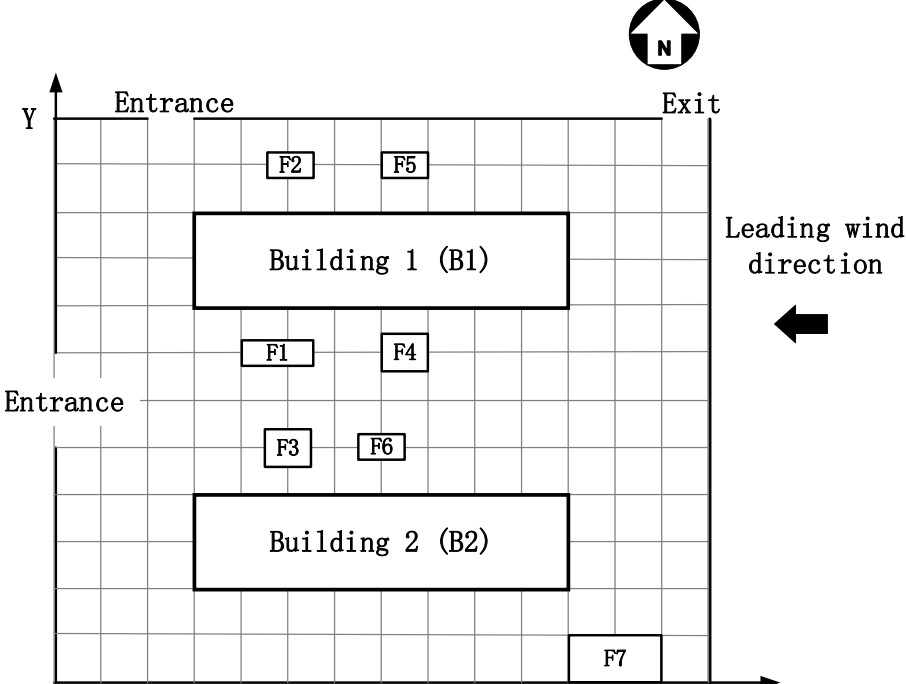

**Figure 6.** Original construction site layout plan.

**Table 4.** Comparison of the construction dust impact and total transportation cost.

| Construction Site Layout Plan | Objective Functions | |
|---|---|---|
| | F1 (mg/m³) | F2 (yuan) |
| Original CSLP | 5.46 | 57,129.62 |
| Optimized CSLP | 2.15 | 31,819.56 |

### 3.3. Result Analysis

It can be seen that the optimized CSLP (see Figure 5) has less construction dust impact and less transportation cost simultaneously compared with the original one. In Figure 5, the temporary facilities are relatively concentrated between two buildings and close to each other. This kind of construction layout could greatly shorten the transportation distance and reduce the total transportation cost. For the templates required on the site, workers need to take them out from storage yard F1, and then transport them to the processing yard F4. After the cutting and other processing processes, templates are finally transported to the floor zone B1 & B2. Since F4 has produced the most serious construction dust pollution among all facilities, it is assigned to the westernmost part of the construction site, close to the site entrance. No other facilities are assigned on the west side of F4. This layout could significantly reduce the construction dust impact of F4 on workers at the downwind area. However, the transportation distance of templates is relatively long. For the rebars, they need to be transported from F2 to F5 for cutting and bending, and then be delivered to two floor zones. F2 and F5 are adjacent to each other and F5 is also close to the two buildings. Therefore, the total transportation distance of rebars is very short. F5 is just adjacent to the

east side of the facility. However, the construction concentration of F5 is relatively low and it would slightly affect the workers' work in F4, B1 and B2.

As for the concrete, the transportation route is from F3 to F6, then to B1 and B2 separately. Among all the building materials, the quantity of concrete that needs to be transported is the largest. Therefore, the layout of F3 and F6 would have a relatively large impact on the total transportation cost. In the optimized plan, F3 and F6 are both located on the middle part of two buildings. The distance between F3 and F6 is also short. This kind of layout could greatly reduce the transportation cost of concrete. Although the dust concentration of concrete mixing station F6 is relatively high, F4 is relatively far away from F6. In other word, F6 does not have much negative impact on workers in F4. However, the workers in F5, B1 and B2 may suffer a certain degree of construction dust. As for the site office F7, it is allocated on the easternmost side of the construction site. Therefore, the construction site staff in F7 would not be affected by construction dust. The optimized CSLP balances the construction dust effect and total transportation cost. The average dust concentration to which workers are exposed is 2.15 mg/m$^3$ and the total transportation cost is CNY 31,819.56.

The original plan (see Figure 6) was designed by site managers, who considered the convenience of transportation and construction on the site. The assignment of facilities in original plan is more widely dispersed than that in optimized CSLP. Due to the sufficient operation space around the facilities, workers can load and unload building materials more conveniently. The sufficient distance between facilities can also ensure the safety and convenience of transportation. However, this layout has some disadvantages. First, the transportation distance of building materials on the site is increased due to this dispersed layout. The total transportation cost is correspondingly increased. According to the results in Table 4, the total transportation cost is increased by 44.3% compared with the optimized plan. Another disadvantage is that site managers did not consider the negative impact of construction dust on workers when planning the site layout. In original plan, storage facilities are all assigned on the western part of the construction site, and the processing facilities are assigned on the eastern part. The storage facilities (F1, F2, F3) are close to the entrances, which is convenient for trunks to unload building materials. However, dust pollution sources (F4, F5, F6) are assigned upwind, and this layout exposes workers downwind to more dust pollution. According to the results, the average construction dust concentration to which workers are exposed is increased by 60.62% compared with the optimized CSLP. In summary, the optimized site layout plan is a more appropriate and reasonable choice for site managers.

## 4. Discussion

### 4.1. Theoretical Implications

This study enriches the dust suppression approaches in the construction industry by incorporating the CSLP method. There are some theoretical implications as follows.

Firstly, this study enriches the research about the construction dust suppression. In previous studies, many researchers have performed many studies about the health impact of PM exposure on workers [3,18]. Based on the research about the health impact assessment, dust monitoring [4,5], this study proposes a new dust suppression method that utilizes the CSLP method to reduce the average dust concentration exposed to workers. Compared with other common dust suppression methods in the construction and other industries, such as local exhaust ventilation (LEV), blower fans, wet methods, and dust masks [3,13,43,44], this method has many advantages. This method does not require any other special equipment and electricity power, which saves much cost and improves the sustainability of construction projects. Additionally, the CSLP method also has a good dust suppression effect. This method reduces construction dust pollution in a more economical, practical and convenient way. In addition, it also enriches the CSLP research and expands the application of the CSLP research.

Secondly, this study develops the relationship between construction dust impact and facilities layout quantitatively. The construction dust-induced occupational health risk is positively correlated with the dust concentration [4]. Therefore, the dust concentration is used as the indicator to assess the occupational health risk. The up–down wind direction method [5,18] was applied to monitor the dust concentration data on the site. With the exponential law model [36], which is suitable for describing the spatial diffusion of construction dust, the quantitative relationship between the construction dust impact and facilities layout could be derived. The quantitative relationship can reveal the effect of the site layout on construction dust pollution exposed to workers and it aids future researchers in searching for more sustainable site layouts.

Thirdly, the proposed model could reduce the construction dust impact and total transportation simultaneously. There is a conflict between these two optimization objectives. In order to reduce the negative impact of construction dust on workers' health, the facilities that generate dust need to be placed on the most downwind position of the construction site [20]. Other facilities should be assigned as far away from them as possible to avoid the potential negative effect. However, this decentralized site layout plan will greatly increase the transportation distance on the site. Therefore, how to balance these two conflicting optimization objectives is a key problem for site managers when planning the site layout. This study quantifies the relationship between the impact of construction dust, transportation cost and site layout, respectively. The MOPSO algorithm is also used to trade off the two objectives and generates a balanced and reasonable construction site layout plan. Therefore, the proposed CSLP method could solve the construction dust pollution problem in a more comprehensive approach.

### 4.2. Practical Implications

In this study, the proposed CSLP method provides a practical, convenient and economical tool for site managers to reduce construction dust pollution and total transportation cost. The practical implications are as follows.

Firstly, those facilities with many workers and little dust, such as the site office, should be assigned to the upwind positions or keep a safe distance from the dust pollution sources. Generally, those facilities with high dust pollution level would pollute the air quality of downwind locations [18]. Therefore, the facilities with many workers should be assigned at upwind locations. If the construction sites cannot be assigned according to the above suggestions, the facilities with many workers should keep a safe distance from dust pollution sources.

Secondly, this method could help site managers to judge the dust pollution level in the pre-construction stage. The dust pollution level of the site layout could be calculated by using the dust diffusion law. For some very small construction sites, the site layouts are difficult to optimize due to the area restriction, and the proposed method cannot be applied directly to these construction sites. Site managers could judge whether to take some other dust suppression methods according to the simulated dust pollution level.

Thirdly, facilities with high interactive flows should be placed close to each other. This kind of site layout could significantly reduce the distance between facilities, and then the total transportation cost can be reduced as well. However, the distance between facilities has a negative impact on dust pollution exposed to workers. In addition, very close distance between facilities would also increase the risk of accidents of workers. Therefore, the site managers could use the proposed method to balance these factors when planning the site layout.

### 4.3. Limitations and Future Directions

One limitation in our research is that the simulation results of dust concentration are influenced by many external factors that are hard to measure. The dust diffusion is a complicated issue [45]. The external factors can affect dust concentration on the site, such as wind speed, relative humidity, and other meteorological factors. Some construction

activities, such as the movement of vehicles, are also positively correlated with the construction dust [46]. However, these influencing factors are irregular. Therefore, it is difficult to monitor and simulate the dust concentration caused by these factors. To mitigate these irregular factors, we suggest that future research should study the relationship between these factors and construction dust in a targeted manner. With these relationships, the generated site layout could be more reasonable.

The other limitation in our research is that the proposed model does not give enough consideration to the convenience of transportation and construction. The move of vehicles and construction activities on site both need enough operation space. In the future, researchers could study the relationship between site layout and construction convenience. Each operation space of construction activity should be determined. The reasonable trunk route should also be planned in the model. With these research, the optimized site layout could be more practical.

## 5. Conclusions

This study presents a BIM-based model to design an optimized site layout plan, which can fulfill the requirements of construction dust negative effect reduction and transportation cost saving, simultaneously. The relationship between construction dust impact and facilities layout can be obtained based on the dust diffusion law and monitoring data of dust concentration on site. The total transportation cost of different layouts can also be simulated according to the BIM model and transportation mode. Finally, the MOPSO algorithm is implemented to search for the optimal site layout plan that can balance the cost and dust pollution effect. The result of the case study indicates that the construction dust pollution effect and transportation cost can be reduced by optimizing the site layout. As the construction dust pollution can bring negative impacts on workers' health, the optimal site layout can provide environmentally friendly site environments for workers by means of dust reduction. Compared with previous research about dust suppression, this study proposes a new method that uses CSLP to reduce dust pollution in a more economical, practical and convenient way. The proposed method can be widely applied in the construction industry. Even for some very small construction sites, it can also help site managers plan the site layouts and improve the occupational health level. With this study, the sustainable and economical objectives can be realized on the construction sites.

**Author Contributions:** Conceptualization, G.T.; methodology, G.T.; software, G.T.; validation, H.F. (Haibo Feng), H.F. (Hui Feng) and K.Z.; formal analysis, H.F. (Haibo Feng); investigation, G.T.; resources, J.F.; data curation, H.F. (Hui Feng); writing—original draft preparation, G.T.; writing—review and editing, H.F. (Haibo Feng) and K.Z.; visualization, G.T.; supervision, J.F. and K.Z.; project administration, J.F.; funding acquisition, J.F. All authors have read and agreed to the published version of the manuscript.

**Funding:** This research was funded by National Social Science Fund of China, grant number 17BGL156; Science and technology project plan of the Ministry of Housing and Urban-Rural Development of the P.R.C., grant number 2018-K8-23.

**Informed Consent Statement:** Informed consent was obtained from all subjects involved in the study.

**Data Availability Statement:** Not applicable.

**Conflicts of Interest:** The authors declare no conflict of interest.

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
