# Peer review of "Reducing Construction Dust Pollution by Planning Construction Site Layout"

_buildings, doi:10.3390/buildings12050531_

Round 1

Reviewer 1 Report

The article shows an interesting issue.
It is undoubtedly a valuable research material.

The strengths of the work include:
- clear introduction to the issues of work and literature review;
- well-presented research methodology;
- graphic presentation of the article: charts, tables, diagrams and drawings which make the manuscript more attractive to the reader.

The weaknesses of the article are:
- the developed method is quite time-consuming and requires an individual approach to a new / different construction;
- the "Discussion" section could be better prepared and include a better references to literature.

Several improvements are required for the manuscript to be published. Below, the authors will find my remarks and comments on the work:

Title

  1. There is a mistake in the title of the work: "Reducing Construction Cust Pollution by Planning Construction Site Layout"

Abstract

  1. The abstract presents an introduction to the subject, research methods, main results and conclusions - however, it exceeds the allowed number of words specified in the journal guidelines. The abstract should be shortened. 

Introduction

  1. Lines: 49, 81-82
    "Many researchers have done research about the health impact of the PM exposure on workers."
    "Many researchers have promoted the dust monitoring research from multiple aspects."

    Such a statement should include a reference to the literature.

  2. Line 50
    "There’re three main research directions." - In this sentence, the research directions should be listed, and then they can be described in detail in the following paragraphs.

Case study

  1. Figure 4
    It would be nice to add a grid of lines to the chart and enlarge it to make the data more readable.

Discussions and Conclusions

  1. The Discussion section and the Conclusions section should be better described.
  2. Construction works are carried out over a long period of time and the wind direction can change significantly during their implementation. Then what?
  3. Sometimes the construction site is very small and it is not possible to apply the methodology presented in the work.
  4. Protecting workers from exposure to dust is a good idea, but it will have a negative effect on other construction parameters, which are usually included in the guidelines, and which are also very important.
  5. Moreover, the compaction of the building site components (workstations, warehouses, etc.) may cause transport difficulties and increase the risk of accidents of workers on the construction site.

Reviewer 2 Report

This is an intriguing subject that appears to make a significant contribution to the field's knowledge, particularly in terms of minimizing the impact of construction activities' pollution on the project's surroundings and, most importantly, on the workers. However, I would like to see the authors address the following issues:

  - The manuscript is undeniably in a severe need of proper editing. Even the article's title contains numerous typos and grammatical errors.   - What is the dust diffusion law, and how does it relate to the layout of the site? - Please justify your selection of tools and methods. - It is unclear how the authors calculated and then reduced the transportation cost on-site using the modeling they used. - What other methods (site planning) have been used previously to reduce such emissions, and what additional benefits can be realized through the implementation of your approach? - Please validate your method by comparing your findings to those of similar studies conducted in other countries and contexts, such as the mining industry.

- Have you considered the workers' movement patterns on the jobsite?

Round 2

Reviewer 2 Report

The authors have addressed the previous comments and the paper seems to have become publishable.